# Comparative Analysis of Secondary Metabolites in *Diplodia corticola* Strains with Different Virulence Degrees Associated with Canker and Dieback of *Quercus* spp.

**DOI:** 10.3390/molecules28176302

**Published:** 2023-08-28

**Authors:** Maria Michela Salvatore, Maria Teresa Russo, Rosario Nicoletti, Alla Eddine Mahamedi, Akila Berraf-Tebbal, Alessio Cimmino, Marco Masi, Anna Andolfi

**Affiliations:** 1Department of Chemical Sciences, University of Naples Federico II, 80126 Naples, Italy; mariamichela.salvatore@unina.it (M.M.S.); mariateresa.russo2@unina.it (M.T.R.); alessio.cimmino@unina.it (A.C.); marco.masi@unina.it (M.M.); 2Institute for Sustainable Plant Protection, National Research Council, 80055 Portici, Italy; 3Department of Agriculture, University of Naples Federico II, 80055 Portici, Italy; rosario.nicoletti@crea.gov.it; 4Council for Agricultural Research and Economics, Research Center for Olive, Fruit and Citrus Crops, 81100 Caserta, Italy; 5Department of Biology, Faculty of Natural Sciences, Life and Earth Sciences, University of Ghardaïa, Ghardaïa 47000, Algeria; aladin1342@yahoo.com; 6Mendeleum-Institute of Genetics and Plant Breeding, Faculty of Horticulture, Mendel University in Brno, 69144 Lednice, Czech Republic; berraf.a@hotmail.fr; 7BAT Center-Interuniversity Center for Studies on Bioinspired Agro-Environmental Technology, University of Naples Federico II, 80055 Portici, Italy

**Keywords:** culture filtrates, mycelial extracts, HPLC-ESI-HRMS, sphaeropsidins, quantitative analysis, virulence factors

## Abstract

*Diplodia corticola* is one of the most aggressive fungal pathogens of *Quercus* species involved in the decline of Mediterranean oak forests. In this study, three strains of *D. corticola* associated with holm (*Quercus ilex*) and cork (*Quercus suber*) oak trees exhibiting dieback symptoms and cankers in Algeria were selected to investigate the production of secondary metabolites. Metabolomic analyses revealed the production of several known compounds, such as sphaeropsidins, diplopyrones and diplofuranones. Moreover, the comparative investigation of secondary metabolites produced by the analyzed strains with different degrees of virulence revealed possible implications of these compounds in the fungal virulence. In particular, sphaeropsidins seem to be the main phytotoxic compounds of *D. corticola* involved in the infections of *Quercus* species, with a possible synergistic influence of the less representative compounds in the fungal virulence.

## 1. Introduction

Oaks (*Quercus* spp.) are among the most economically and ecologically important broad-leaved forest tree species in the Mediterranean region [1]. In the last few decades, oak forests have been seriously affected by decline phenomena, causing global and local concerns. In fact, many studies describe the poor health of oak trees in many forests of the world [2,3,4,5,6,7,8,9]. The oak dieback or decline can be caused by a range of factors, including extreme climatic events, insects, and pathogenic fungi [10,11,12,13].

The involvement of fungi from the family Botryosphaeriaceae in oak decline has been well documented [14,15]. In particular, *Botryosphaeria dothidea* (Moug.) Ces. & De Not., *Diplodia corticola* A.J.L. Phillips, A. Alves & J. Luque, *Diplodia seriata* De Not., *Dothiorella sarmentorum* (Fr.) A.J.L. Phillips, Alves & Luque, and *Neofusicoccum parvum* (Pennycook & Samuels) Crous, Slippers & A.J.L. Phillips were isolated with high frequency from physiologically impaired trees [15,16,17,18]. The phenomenon turned particularly severe in those geographic areas that are more exposed to global warming effects [19]. Main symptoms of disease include combinations of acute wilt, blighted leaves, external cankers, necrotic lesions and severe dieback [17]. Among the botryosphaeriaceous fungi [20]*, D. corticola* is able to infect a wide range of *Quercus* species, with special incidence in Mediterranean taxa, such as *Quercus afares* Pomel, *Quercus canariensis* Willd., *Quercus coccifera* L., *Quercus ilex* L. and *Quercus suber* L. [21]. Additionally, it has been cited as an emerging pathogen in American species of oaks [17] (Figure 1). During a study on Botryosphaeriaceae species associated with oak decline in Tunisia, Linaldeddu et al. reported the existence of two distinct lineages within *D. corticola* (i.e., A and B) [22,23]. *Diplodia corticola* is able to produce many bioactive secondary metabolites, which could be involved as virulence factors in host–pathogen interactions [24,25,26,27].

Considering that virulence of *D. corticola* is strain dependent [28], we selected three strains of *D. corticola* associated with holm (*Q. ilex*) and cork (*Q. suber*) oak trees exhibiting dieback symptoms and cankers in Algeria [15] to investigate the production of secondary metabolites. In fact, we hypothesized that different virulence degrees of strains of *D. corticola* are associated with differences in the production of secondary metabolites. For this reason, the metabolic profiles of strains MAEC02 and MAEC10 associated with cork oak from lineage A and B, respectively, were compared due to their different virulence degree. Moreover, the production of secondary metabolites was also investigated for the strain MAEC03 (lineage A) of *D. corticola* isolated from the holm oak, in order to reveal possible metabolic differences related to the host and not only to the strain virulence. Hence, this paper reports the results of metabolomic analyses of crude extracts obtained from culture filtrates and mycelia of strains MAEC02, MAEC03 and MAEC10 of *D. corticola,* performed essentially through NMR, HPLC-UV and HPLC-ESI-HRMS.

## 2. Results

### 2.1. Pathogenicity Trials

Strains MAEC02, MAEC03 and MAEC10 of *D. corticola* were pathogenic on detached green shoots of *Q. suber* and *Q. ilex,* causing brown to black colored lesions and progressing upward and downward from the inoculation point. No lesions developed in control plants. In addition, all the strains were successfully re-isolated from the necrotic wood.

Strains show variable aggressiveness, as expressed by the lesion lengths they caused on their hosts (Figure 2). *Diplodia corticola* MAEC10 is the most aggressive strain on *Quercus* shoots.

### 2.2. Qualitative and Quantitative Analyses of Secondary Metabolites

Crude extracts of culture filtrates identified, respectively, as CE02, CE03 and CE10, and crude mycelial extracts identified, respectively, as ME02, ME03 and ME10, were obtained from strains MAEC02, MAEC03 and MAEC10 of *D. corticola*. All the cultures were prepared in triplicate.

In the current study, mass spectrometry (i.e., HPLC-ESI-HRMS), chromatographic (i.e., TLC, HPLC-UV) and spectroscopic (i.e., NMR) techniques were applied to determine the metabolic composition of the crude extracts.

Preliminary thin layer chromatography (TLC) data showed the presence, in all the investigated crude extracts, of two intense spots which were attributed to sphaeropsidins A and B (Sph A and Sph B) by using the comparison with pure standards resulting from our previous studies [25,27]. Further NMR investigations confirmed the presence of Sph A and Sph B in the crude extracts. In fact, the ^1^H NMR spectrum of CE10 (Appendix A) essentially shows the typical signals pattern of Sph A and Sph B [29,30], while ^1^H NMR spectra of CE02 and CE03 seem to be more complex, due to the presence of additional signals that could be attributed to other metabolites. Moreover, NMR spectra of ME02, ME03 and ME10 (Appendix A) show signals attributable to hydroxy fatty acids [31].

The putative presence of other metabolites in the crude extracts, which can be deduced by examining the complexity of acquired ^1^H NMR spectra, triggered us to conduct a deeper investigation of the samples via HPLC-ESI-HRMS (Appendix A). In fact, secondary metabolite detection was confirmed through extracted ion chromatograms, and signals in the acquired high-resolution mass spectra of main chromatographic peaks (which correspond essentially to adducts with hydrogen, sodium, potassium) are reported in Table 1 and Appendix A. Moreover, dimer and peaks generated after the loss of H_2_O or OH from parent ion were detected.

The identities of the compounds reported in Table 1 were confirmed by comparison with analytical standard compounds. As it is known, in the case of stereoisomers or isomers with minimal structural differences, HPLC-ESI-HRMS is not able to make an unambiguous identification and even the use of analytical standards does not help to confirm the molecular structures. For this reason, more than one compound is considered for some entries in Table 1.

The structures of the secondary metabolites detected in cultural and mycelial extracts of MAEC02, MAEC03 and MAEC10 strains of *D. corticola* are presented in Figure 3.

The occurrence of secondary metabolites in the culture filtrate and mycelial extracts is shown in Table 2. Interestingly, Sphs A, B and C were detected in all the investigated crude extracts.

Considering that Sph A and its analogues are already known to be the main phytotoxins produced by *Diplodia* species [24], we conjectured that these members of the family of tetracyclic pimarane diterpenes could also be involved in the different virulence degrees observed in the strains of *D. corticola* from oaks investigated in this study. For this reason, Sphs A and B were selected to evaluate quantitative differences among the strains.

Hence, for an accurate quantification of Sphs A and B, a HPLC-UV method was optimized for the best peak resolution and minimum run time using analytical standard compounds. To this end, the External Standard Calibration Method (ESTD) was employed as reported in Section 4.

Figure 4 shows the concentrations of Sphs A and B in culture filtrates of *D. corticola* MAEC02, MAEC03 and MAEC10. As can be seen, *D. corticola* MAEC10, isolated from cork oak and the most virulent strain, appears to be the highest Sph A producer, while similar concentrations of Sph B were detected in the culture filtrates of both lineages of *D. corticola* from cork oak. Our data also show that the culture filtrates of *D. corticola* MAEC03 isolated from holm oak exhibit lower concentrations of Sphs A and B than the culture filtrates of the strains isolated from cork oak. Hence, among the strains of *D. corticola* investigated in this study, the most virulent one (i.e., MAEC10) appears to also be the strain that produces most sphaeropsidins.

## 3. Discussion

*Diplodia corticola* is responsible for serious diseases in forest trees in the Mediterranean region. In particular, this fungus is one of the most aggressive pathogens of *Quercus* species and it is involved in the decline of oak forests [28].

It is known that fungal secondary metabolites play crucial roles in pathogenic interactions as virulence factors [26,32,33,34,35]. Indeed, there are several reports that these products directly contribute to the outcome of infections [36,37,38]. In this respect, Sph A, a tetracyclic pimarane diterpene, provides an excellent example due to its proven role in cypress cankers induced by *Diplodia cupressi* A.J.L. Phillips & A. Alves [24]. In fact, when tested for phytotoxicity, Sph A induced yellowing, browning and necrosis on *Cupressus sempervirens* L., dieback on *Cupressus macrocarpa* Hartweg. ex Gordon, and browning and necrosis on *Cupressus arizonica* Greene. Furthermore, herbaceous plants are also affected by this phytotoxin that causes yellowing and necrosis on *Avena sativa* L. and epinasty and necrosis on *Lycospersicon esculentum* Mill. [39].

In the present study, pathogenicity of strains of *D. corticola* isolated from cork and holm oaks was tested on symptomless shoots of *Q. ilex* and *Q. suber.* These trials showed the ability of all the tested strains to produce brown/black colored lesions. As expressed by the lesion lengths they caused on oaks (Figure 2), the strains under investigation have different degrees of virulence. *Diplodia corticola* MAEC10 turned out to be the most aggressive strain on *Quercus* shoots. These strains were also investigated for the production of secondary metabolites, in order to reveal possible mediators of fungal virulence. A variety of secondary metabolites was identified in culture filtrate and mycelial extracts of *D. corticola* MAEC02, MAEC03 and MAEC10. From the comparative study of the fungal products, a different secondary metabolites profile was observed for each investigated strain (Table 2). Sph A and two analogues (i.e., Sphs B and C) were identified in all the culture filtrate and mycelial extracts under examination. In addition to phytotoxicity, Sph A has a broad array of biological activities such as antimicrobial, insecticidal, herbicidal and anticancer [24,25,40,41]. Sphs B and C have also been tested for phytotoxicity on non-host plants, like tomato and oat, inducing necrosis on cuttings and brown discoloration or stewing on the stem [30]. Therefore, the detection of Sphs turns out to be particularly relevant. The quantification of Sphs A and B produced by MAEC02, MAEC03 and MAEC10 shows that the most virulent strain (i.e., MAEC10) is also the strain that produces most Sphs (Figure 4), proving that these compounds play a crucial role in the fungal infection.

Besides sphaeropsidins, HPLC-ESI-MS data suggest the presence in the crude extracts of other metabolites already known as products of *Diplodia* species (e.g., diplopyrones, diplobifuranylones, diplofuranones) (Table 2). However, some metabolites were produced by only one of the strains of *D. corticola* investigated in this study. This is the case of diplopyrone B, a phytotoxic α-pyrone for cork oak, grapevine and tomato [27], which was exclusively detected as a product of the most virulent strain under examination (i.e., MAEC10). Furthermore, several metabolites are present in CE02 and CE03, but not in CE10. However, a number of signals reported in Table 1 could either not be identified or could be explained by more than one compound. For instance, the signal at retention time of 4.24 min was detected in both CE02 and CE03 chromatograms, but the interpretation of the high-resolution mass spectrum corresponding to this chromatographic peak does not allow for discrimination between the diplobifuranylones A and B, because these compounds are two diastereomeric 5′-(1-hydroxyethyl)-3,4,2′,5′-tetrahydro-2*H*-[2,2′]bifuranyl-5-ones [42]. At any rate, diplobifuranylones A and B were previously identified from *D. corticola* isolated from cork oak and both compounds did not show phytotoxic activity when assayed on a non-host plant (tomato) [43].

As expected, hydroxylinolenic acid and hydroxylinoleic acid were exclusively detected in mycelial extracts because fatty acids and their derivatives are well-known constituents of the fungal cell membranes [44]. Moreover, several “compounds” not identified (listed as “Unknown” in Table 2) were only detected in the mycelial extracts (e.g., Unknown5, Unknown8, Unknown12). Their exclusive presence in mycelial extracts suggests their implication in the biomass formation of the microbial communities. However, since their mass spectra cannot be linked to specific substances, any consideration of their biological role is hampered.

## 4. Materials and Methods

### 4.1. Fungal Strains and Fermentation Conditions

*Diplodia corticola* strains were isolated from holm and cork oak trees exhibiting dieback symptoms and cankers in Algeria. The strains were identified using multi-gene phylogenetic analysis, based on DNA sequence data of the internal transcribed spacer region (ITS) and translation elongation factor (*tef1-α*) [15]. Pure cultures of hyphal tipped isolates were maintained in the culture collection of the Mendeleum-Institute of Genetics and Plant Breeding, Faculty of Horticulture, Mendel University in Brno, Lednice, Czech Republic. Stock cultures of selected strains were maintained on potato dextrose agar (Oxoid, Ottawa, ON, Canada) and mycelial plugs were used to inoculate 250 mL of Czapek Dox broth (Oxoid), added with 2% yeast extract in 500 mL Erlenmeyer flasks. The cultures were incubated in the stationary phase in the dark at 25 °C. After 30 days, the liquid phase was separated by using filtration on Whatman No. 5 filter paper, and the culture filtrates and respective mycelia were stored at −20 °C. The experiments were carried out in triplicate.

### 4.2. Pathogenicity Trials

Pathogenicity of *D. corticola* strains was tested on symptomless detached shoots of *Q. ilex* and *Q. suber* as previously described [15]. Briefly, small strips of bark (<5 mm) were aseptically cut from the shoots, where a potato dextrose agar (PDA) plug with mycelial growth was deposited. Sterile damp cotton and Parafilm were wrapped around the inoculation point to avoid desiccation. Five replicates were used for each tested strain. In addition, five asymptomatic shoots were inoculated with sterile PDA plugs and were used as control. All inoculated shoots were watered and incubated at room temperature, under a daily photoperiod. Forty days later, the bark was removed and lesion lengths around the inoculation point were measured. Koch’s postulates were fulfilled after re-isolating the inoculated fungi in PDA.

### 4.3. Extracts Procedures

Three replicates of culture filtrates (≈220 mL) of *D. corticola* MAEC02, MAEC03 and MAEC10 were extracted three times with ethyl acetate (EtOAc) at native pH (= 6). Organic phases were dried with anhydrous Na_2_SO_4_ and evaporated under reduced pressure, yielding culture filtrate crude extracts (CE) as brownish solid residues (CE02 = 85 ± 8 mg, CE03 = 35 ± 6 mg, CE10 = 110 ± 9 mg).

Mycelia of each strain (three replicates) were extracted with 150 mL of MeOH-H_2_O (NaCl 1%) mixture (55:45, *v*/*v*) and homogenized in a mixer. A supernatant was obtained after centrifugation (30 min at 8000 rpm, 8 °C). The residue was further subjected to a second homogenization with 100 mL of the mixture reported above, followed by centrifugation. The two supernatants were combined and concentrated under reduced pressure to remove methanol. The obtained aqueous phase was extracted (three times) with EtOAc at native pH (=6). The organic phases were combined, dried on anhydrous Na_2_SO_4_, and evaporated under reduced pressure yielding mycelial crude extract (ME) as brownish solids (ME02 = 55 ± 4 mg, ME03 = 50 ± 3 mg, ME10 = 50 ± 4 mg).

^1^H NMR spectra were recorded on a Bruker AMX instrument at 400 MHz in CDCl_3_. The same solvents were used as internal standard. TLC were performed on silica gel (Kieselgel 60, F_254_, 0.25 mm, Merck, Darmstadt, Germany) eluted with CHCl_3_/*iso*-PrOH (95:5 *v*/*v*; SphA and Sph B R_f_ 0.50 and 0.37, respectively). The spots were visualized by exposure to UV radiation (253 nm), or by spraying them first with 10% H_2_SO_4_ in methanol, followed by heating at 110 °C for 10 min.

### 4.4. Qualitative Analysis

The culture filtrates and mycelia extracts (CEs and MEs) were analyzed by an Agilent (Cernusco sul Naviglio, Milan, Italy) HPLC-MS ESI-TOF 1260/6230DA instrument operating in positive ionization mode. The source temperature was kept at 120 °C and the desolvating gas at 250 °C. The instruments were interfaced to a Phenomenex Luna 5µ C18 column (150 × 4.6 mm, 5 µm). An acetonitrile-water (0.1% formic acid) gradient was used starting from 30% acetonitrile, increasing linearly to 60% in 30 min, 60% for 10 min and finally rebalanced to the initial rate for 5 min. Methanol was used to dissolve the samples. HPLC flow rate was 500 µL min^−1^.

Pure compounds employed as analytical standards for the qualitative analysis were obtained as previously reported [25,27].

### 4.5. Quantitative Analysis

The quantification of Sphs A and B in crude extracts of *D. corticola* MAEC02, MAEC03 and MAEC10 was carried out via Hitachi (Tokyo, Japan) HPLC system paired with a 5160 pump and a 5410 spectrophotometric detector. A Phenomenex Luna 5µ C18 column (150 × 4.6 mm, 5 µm) was employed for the analysis, using the same phase mobile gradient and conditions detailed in Section 4.4. The detection was performed at 220 or 240 nm.

The quantification was performed with a five points calibration curve in the range 500–31.25 mg L^−1^ for Sph A and 2000–125 mg L^−1^ for Sph B. Sphs A and B employed as analytical standards for the quantitative analysis were obtained as previously reported [25]. First, weighted amounts of standard compounds were dissolved in methanol to obtain calibration solutions containing accurately known concentrations of the target analyte. For each target analyte, a calibration plot was constructed by reporting (at each calibration level) the integrated area of the chromatographic peak as a function of the known concentration of the analyte.

Visual inspection of each calibration plot readily showed a linear dependence of the signal from the concentration. In fact, calibration data could be perfectly interpolated with a least square regression line (*y* = *a* + *bx*) in which *a* is the intercept and *b* the slope (L mg^−1^) of the least square line through the calibration points. This can be deduced from the determination coefficients (R^2^) of Sphs A and B reported in Table 3 which are very close to one.

Table 3 also reports the intercepts and slopes deduced from the two calibration plots together with the corresponding standard deviations (respectively, $s_{a}$ and $s_{b}$) which were calculated from the calibration data according to standard statistical procedures [45].

The standard deviation of the intercept, $s_{a}$, is used to calculate the method detection limit (LOD) according to the formula: $\mathrm{LOD}=3.3\times s_{a}/b$; and the quantification limit (LOQ) according to the formula: $\mathrm{LOQ}=10\times s_{a}/b$.

The concentrations and standard deviations of Sphs A and B determined in each crude extract were converted to the corresponding concentrations and standard deviations in each culture filtrate (see Figure 4).

### 4.6. Statisical Analysis

Statistical analyses were performed using MATLAB R2022b (Mathworks, Natick, MA, USA). Data were reported as mean of independent experiments ± standard deviation. Analysis of variance (ANOVA) was applied, followed by a Tukey multiple comparison test, and values of *p*-value *<* 0.05 were considered statistically different.

## 5. Conclusions

*Diplodia corticola* is responsible of serious damages of oak forests in the Mediterranean regions. It is also known as a rich source of bioactive secondary metabolites, which could be involved in plant–pathogen interactions. In this study, the comparative investigation of secondary metabolites produced by strains of *D. corticola* isolated from cork and holm oaks with different degrees of virulence revealed possible implications of these compounds in the fungal virulence. Our attention was focused on sphaeropsidins which are already known to be the main phytotoxins produced by *D. cupressi*, the causal agent of a severe canker disease in Italian cypresses [24]. Our data showed the different production of sphaeropsidins A and B among different strains of *D. corticola* and, in particular, we observed that the most virulent strain investigated in this work (i.e., MAEC10) is also the strain that produces most sphaeropsidins. Thus, this evidence suggests a possible role of such important phytotoxins in the virulence of *D. corticola*. However, the further detected metabolites may be considered for a possible synergistic effect in the fungal virulence.

To fully understand the infection mechanism and the oak decline, future research should aim to directly assess the production of these fungal secondary metabolites in infected tissues of *Quercus* spp. in order to establish whether or not the metabolic differences observed in vitro correspond to the biosynthetic levels in planta.

## Figures and Tables

**Figure 1 molecules-28-06302-f001:**
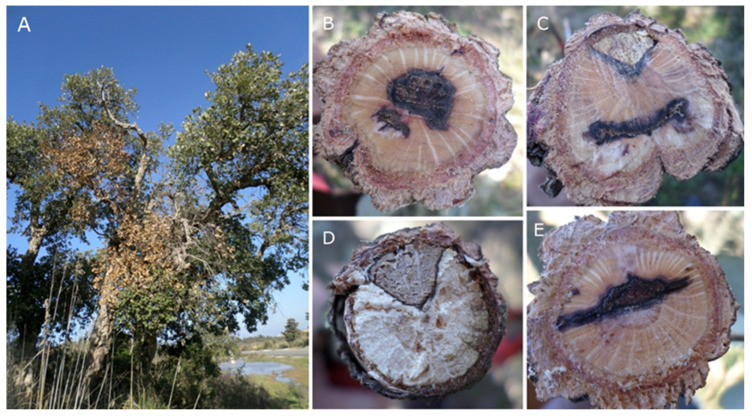
Cork oak tree showing extensive twig and branch dieback (**A**); internal lesions of sectioned branches (**B**–**E**).

**Figure 2 molecules-28-06302-f002:**
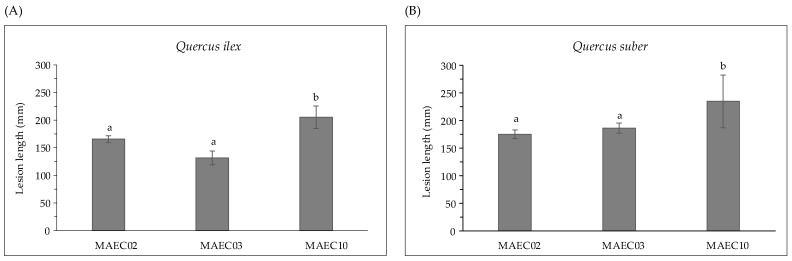
Lesion lengths (mm) caused by MAEC02, MAEC03 and MAEC10 strains of *Diplodia corticola* inoculated on detached green shoots of (**A**) *Quercus ilex* and (**B**) *Quercus suber*. Tests were performed in five replicates and results are presented as the mean ± standard deviation (SD). Data with different letters (a,b) in each panel are significantly different (one-way ANOVA followed by Tukey’s post hoc test; *p* < 0.05). Values with dissimilar letters are significantly different from each other (*p* < 0.05). Values with the same letter are not significantly different (*p* > 0.05).

**Figure 3 molecules-28-06302-f003:**
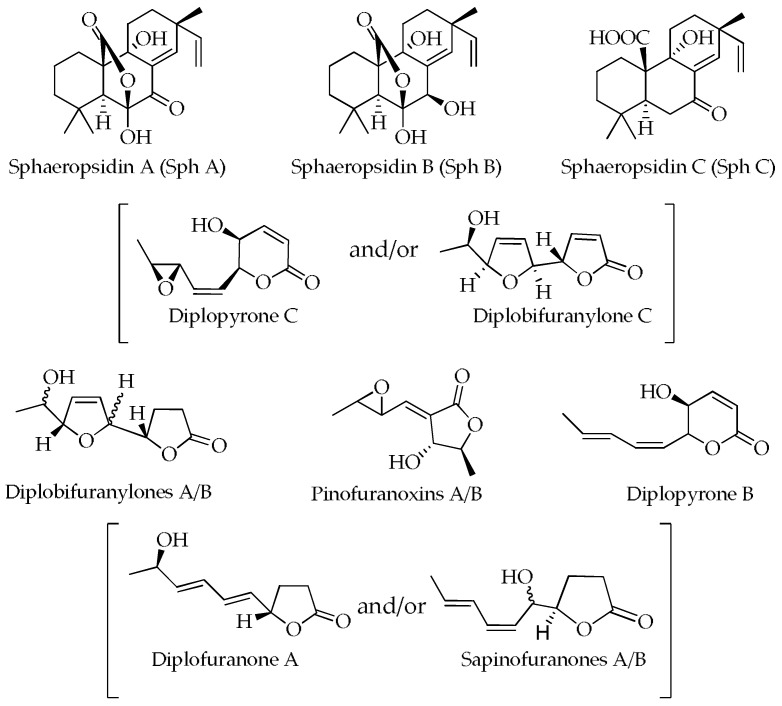
Structures of metabolites detected in the culture filtrate (CE) and mycelial (ME) extracts of MAEC02, MAEC03 and MAEC10 strains of *Diplodia corticola*.

**Figure 4 molecules-28-06302-f004:**
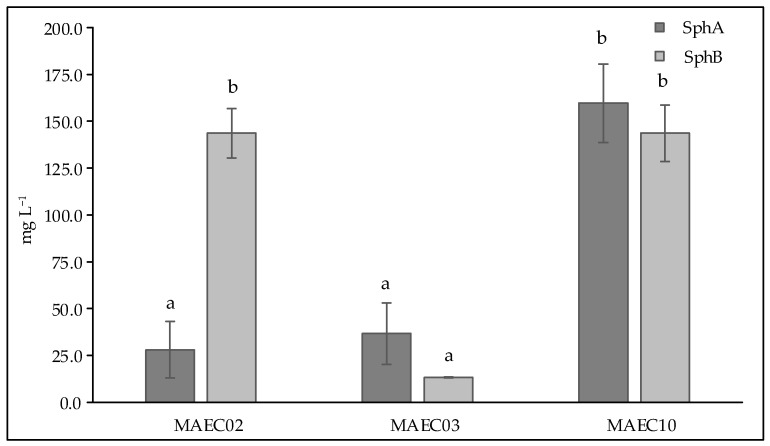
Concentrations (expressed as mg L^−1^) of Sph A and Sph B in culture filtrates of *Diplodia corticola* MAEC02, MAEC03 and MAEC10. Analyses were performed in triplicate and results are presented as the mean ± standard deviation. Data with different letters (a,b) are significantly different (two-way ANOVA followed by Tukey’s post hoc test; *p* < 0.05). Values with dissimilar letters are significantly different from each other (*p* < 0.05). Values with the same letter are not significantly different (*p* > 0.05).

**Table 1 molecules-28-06302-t001:** HPLC-ESI-HRMS analysis of cultural and mycelial extracts of MAEC02, MAEC03 and MAEC10 strains of *Diplodia corticola*.

Compound	Rt (Min)	Formula	Exact Mass (U) ^1^	Diagnostic Ions *m*/*z* (Calcd. [M + H]^+^ or [M + Na]^+^)
Unknown1	3.16	-	216	455.1843 [2M + Na]^+^, 255.0631 [M + K]^+^, 239.0879 [M + Na]^+^, 217.1052 [M + H]^+^
Unknown2	3.26	-	214	451.1557 [2M + Na]^+^, 429.1734 [2M + H]^+^, 253.0468 [M + K]^+^, 237.0735 [M + Na]^+^, 215.0915 [M + H]^+^, 197.0808 [M − OH]^+^, 179.0704 [M − OH − H_2_O]^+^
Diplobifuranylones A/B ^2^	4.24	C_10_H_14_O_4_	198.0892	435.1580 [2M + K]^+^, 419.1661 [2M + Na]^+^, 397.1838 [2M + H]^+^, 237.0535 [M + K]^+^, 221.0784 [M + Na]^+^, 199.0965 [M + H]^+^ (199.0970), 181.0855 [M − OH]^+^, 155.0706 [M + H − CO_2_]^+^
Diplopyrone C and/or diplobifuranylone C	4.85	C_10_H_12_O_4_	196.0736	415.1359 [2M + Na]^+^, 235.0392 [M + K]^+^, 219.0627 [M + Na]^+^, 197.0790 [M + H]^+^ (197.0814), 179.0704 [M − OH]^+^
Pinofuranoxins A/B ^2^	6.39	C_9_H_12_O_4_	184.0736	391.1367 [2M + Na]^+^, 207.0628 [M + Na]^+^, 185.0810 [M + H]^+^ (185.0814)
Unknown3	7.61	-	168	359.1488 [2M + Na]^+^, 191.0679 [M + Na]^+^, 169.0860 [M + H]^+^
Diplofuranone A and/or sapinofuranones A/B	9.05	C_10_H_14_O_3_	182.0943	387.1752 [2M + Na]^+^, 221.0560 [M + K]^+^, 205.0831 [M + Na]^+^, 183.1026 [M + H]^+^ (183.1021), 165.0908 [M − OH]^+^
Unknown4	9.86	-	350	723.4011 [2M + Na]^+^, 389.1783 [M + K]^+^, 373.1975 [M + Na]^+^, 333.2036 [M − OH]^+^
Diplopyrone B	13.01	C_10_H_12_O_3_	180.0786	383.1443 [2M + Na]^+^, 219.0405 [M + K]^+^, 203.0678 [M + Na]^+^, 181.0857 [M + H]^+^ (181.0865), 163.0756 [M − OH]^+^
Unknown5	15.63	-	310	643.4107 [2M + Na]^+^, 349.1797 [M + K]^+^, 333.2010 [M + Na]^+^, 293.2083 [M − OH]^+^
Unknown6	17.65	-	314	651.2913 [2M + Na]^+^, 357.2020 [M + K]^+^, 337.1404 [M + Na]^+^, 297.1480 [M − OH]^+^
Unknown7	20.87	-	320	663.4526 [2M + Na]^+^, 359.2082 [M + K]^+^, 343.2220 [M + Na]^+^, 303.2299 [M − OH]^+^, 285.2197 [M – OH − H_2_O]^+^
Sph C	22.25	C_20_H_28_O_4_	332.1988	1019.5783 [3M + Na]^+^, 687.3826 [2M + Na]^+^, 665.4014 [2M + H]^+^, 355.1871 [M + Na]^+^, 333.2056 [M + H]^+^ (333.2066), 315.1947 [M − OH]^+^
Sph B	24.01	C_20_H_28_O_5_	348.1937	735.3452 [2M + K]^+^, 719.3727 [2M + Na]^+^, 387.1557 [M + K]^+^, 371.1819 [M + Na]^+^, 349.2001 [M + H]^+^ (349.2015), 331.1897 [M − OH]^+^, 313.1794 [M – OH − H_2_O]^+^
Unknown8	25.80	-	312	663.3997 [2M + K]^+^, 647.4413 [2M + Na]^+^, 625.4603 [2M + H]^+^, 351.1787 [M + K]^+^, 335.2167 [M + Na]^+^, 295.2249 [M − OH]^+^, 277.2143 [M – OH − H_2_O]^+^
Unknown9	26.16	-	328	695.3066 [2M + K]^+^, 679.3191 [2M + Na]^+^, 367.1393 [M + K]^+^, 351.1557 [M + Na]^+^, 311.1631 [M − OH]^+^
Unknown10	29.96	-	320	679.4442 [2M + K]^+^, 663.4514 [2M + Na]^+^, 359.2104 [M + K]^+^, 343.2217 [M + Na]^+^, 303.2289 [M − OH]^+^, 285.2192 [M – OH − H_2_O]^+^
Sph A	32.30	C_20_H_26_O_5_	346.1780	715.3422 [2M + Na]^+^, 385.1400 [M + K]^+^, 369.1672 [M + Na]^+^, 347.1852 [M + H]^+^ (347.1858), 329.1745 [M − OH]^+^
Unknown11	34.79	-	300	623.3314 [2M + Na]^+^, 601.3183 [2M + H]^+^, 323.1614 [M + Na]^+^, 301.1483 [M + H]^+^, 283.1690 [M − OH]^+^
Hydroxylinolenic acid	35.96	C_18_H_30_O_3_	294.2195	627.3961 [2M + K]^+^, 611.4226 [2M + Na]^+^, 589.4403 [2M + H]^+^, 333.1801 [M + K]^+^, 317.2070 [M + Na]^+^ (317.2093), 277.2147 [M − OH]^+^
Unknown12	37.94	-	312	647.4414 [2M + Na]^+^, 351.1883 [M + K]^+^, 335.2171 [M + Na]^+^, 313.2352 [M + H]^+^, 295.2244 [M − OH]^+^, 277.2139 [M − OH − H_2_O]^+^
Hydroxylinoleic acid	40.41	C_18_H_32_O_3_	296.2326	615.4529 [2M + Na]^+^, 335.1944 [M + K]^+^, 319.2223 [M + Na] ^+^ (319.2249), 279.2306 [M − OH]^+^
Unknown13	42.21	-	565	1153.8128 [2M + Na]^+^, 604.3767 [M + K]^+^, 588.4040 [M + Na]^+^, 566.4222 [M + H]^+^
Unknown14	43.41	-	434	473.2463 [M + K]^+^, 457.2792 [M + Na]^+^, 435.2510 [M + H]^+^

^1^ Molecular masses of unknown compounds are approximated; ^2^ These compounds are stereoisomers.

**Table 2 molecules-28-06302-t002:** Distribution of secondary metabolites in culture filtrate (CE) and mycelial (ME) extracts of MAEC02, MAEC03 and MAEC10 strains of *Diplodia corticola* (i.e., CE02, CE03, CE10, ME02, ME03, ME10).

Compound	Rt (Min)	MW	CE02	CE03	CE10	ME02	ME03	ME10
Unknown1	3.16	216	-	-	-	+	+	-
Unknown2	3.26	214	+	+	+	+	-	+
Diplobifuranylones A/B ^1^	4.24	198	+	+	-	+	+	-
Diplopyrone C and/or diplobifuranylone C	4.85	196	+	+	-	+	-	-
Pinofuranoxins A/B ^1^	6.39	184	+	-	-	-	-	-
Unknown3	7.61	168	+	-	-	-	-	-
Diplofuranone A and/or Sapinofuranones A/B	9.05	182	+	+	-	+	-	-
Unknown4	9.86	350	-	-	+	-	-	+
Diplopyrone B	13.01	180	-	-	+	-	-	+
Unknown5	15.63	310	-	-	-	+	+	+
Unknown6	17.65	314	-	-	+	-	-	+
Unknown7	20.87	320	-	-	-	-	-	+
Sph C	22.25	332	+	+	+	+	+	+
Sph B	24.01	348	+	+	+	+	+	+
Unknown8	25.80	312	-	-	-	+	+	+
Unknown9	26.16	328	-	-	+	-	-	-
Unknown10	29.96	320	-	-	-	-	-	+
Sph A	32.30	346	+	+	+	+	+	+
Unknown11	34.79	300	-	-	+	-	-	-
Hydroxylinolenic acid	35.96	294	-	-	-	+	+	+
Unknown12	37.94	312	-	-	-	-	+	+
Hydroxylinoleic acid	40.41	296	-	-	-	+	+	+
Unknown13	42.21	565	-	-	-	+	+	+
Unknown14	43.41	434	-	-	-	+	+	-

^1^ These compounds are stereoisomers. + detected; - not detected.

**Table 3 molecules-28-06302-t003:** Performance data of the optimized HPLC-UV quantification method.

Metabolite	Wavelength (nm)	Rt (Min)	Intercept $\left(a\pm s_{a}\right)$	Slope $\left(b\pm s_{b}\right)\;L\;mg^{-1}$	R^2^	LOD (mg L^−1^)	LOQ (mg L^−1^)
Sph A	240	32.30	$2.68\times 10^{5}$ ± $\;1.96\times 10^{5}$	$6.14\times 10^{4}$ ± $\;8.30\times 10^{2}$	0.999	13.05	39.5
Sph B	220	24.01	$1.73\times 10^{5}$ ± $\;8.99\times 10^{4}$	$7.15\times 10^{3}$ ± 95.1	0.999	44.23	134.0

## Data Availability

The data that support the findings of this study are available from the corresponding author upon reasonable request.

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
