# Peer review of "Comparative Analysis of Secondary Metabolites in *Diplodia corticola* Strains with Different Virulence Degrees Associated with Canker and Dieback of *Quercus* spp."

_molecules, 2023, doi:10.3390/molecules28176302_

Round 1

Reviewer 1 Report

Comments to manuscript: “Comparative Analysis of Secondary Metabolites in Diplodia 2 corticola Strains with Different Virulence Degrees associated 3 with Canker and Dieback of Quercus spp.”

Manuscript ID: 2532776

It is interesting to know the virulence factors that fungi use to cause diseases in other living beings. The results on the metabolites produced by Diplodia can provide information to understand the evolution of diseases in plants.

Could the authors explain if they tried to extract metabolites from infected plant tissue in this work and thus compare these results with those obtained in fungal cultures? Please include your explanation in discussions.

Otherwise, I have no comments.

Author Response

Comments to manuscript: “Comparative Analysis of Secondary Metabolites in Diplodia 2 corticola Strains with Different Virulence Degrees associated 3 with Canker and Dieback of Quercus spp.”

Manuscript ID: 2532776

 It is interesting to know the virulence factors that fungi use to cause diseases in other living beings. The results on the metabolites produced by Diplodia can provide information to understand the evolution of diseases in plants.

Could the authors explain if they tried to extract metabolites from infected plant tissue in this work and thus compare these results with those obtained in fungal cultures? Please include your explanation in discussions.

We agree with the referee on the importance of evaluating metabolites in the infected plant tissues and it could be an interesting idea for a future work. So we added it as a future aim in the conclusion section. (see Lines 329-333)

 Otherwise, I have no comments.

We wish to thank the referee for the positive comments.

Reviewer 2 Report

Current manuscript entitled, "Comparative Analysis of Secondary Metabolites in Diplodia corticola Strains with Different Virulence Degrees associated with Canker and Dieback of Quercus spp.", investigates secondary metabolites produced by three strains of an aggressive pathogen infecting  two Oak  species. They have found sphaeropsidins as the main phytotoxic compounds of D. corticola involved in the infections of Quercus species.

There are too many short paragraphs in the introduction. Related paragraphs should be merged.

Materials and Methods section must be placed before Result section.

An author citation of each scientific name is required where they are used first in the text e.g., line 43, 44, 50, 51, 166,167, 169

line 42, 53: each rank of classification must be italicized Botryosphaeriaceae

line 73: A separate methodology section is required before results.

line 78: reisolated….re-isolated

line 88: All the cultures were performed in triplicate. Sentence needs revision

line 108, 241, 245, 258: H2O, Na2SO4, MeOH-H2O,  H2SO4…. Please correct as all numeric values within the formulae as subscripts

line 219: Mendeleum Institute of Genetics…Is it Mendeleum-Institute of Genetics and Plant Breeding, Brno, Czech Republic?

line 232: an equal number of shoots inoculated with sterile PDA plugs…..state the number of shoots please

table 2 : numerical numbers in the table should be subscripts

conclusion: elaborate and include some future prospects please

Needs English language revision

Author Response

Current manuscript entitled, "Comparative Analysis of Secondary Metabolites in Diplodia corticola Strains with Different Virulence Degrees associated with Canker and Dieback of Quercus spp.", investigates secondary metabolites produced by three strains of an aggressive pathogen infecting  two Oak  species. They have found sphaeropsidins as the main phytotoxic compounds of D. corticola involved in the infections of Quercus species.

There are too many short paragraphs in the introduction. Related paragraphs should be merged.

As suggested, related subparagraphs in the introduction were merged. (see lines 48, 54, 63, 66)

Materials and Methods section must be placed before Result section.

According to the “Instruction for Authors” of the journal “Molecules” (https://www.mdpi.com/journal/molecules/instructions), the manuscript was prepared using the Microsoft Word Template in which the Results section is placed right after the Introduction section.

An author citation of each scientific name is required where they are used first in the text e.g., line 43, 44, 50, 51, 166,167, 169

As suggested by the referee, author citations of the scientific name of each plant and fungal species were added.

line 42, 53: each rank of classification must be italicized Botryosphaeriaceae

the term “Botryosphaeriaceae” for editorial reasons is not italicized in the manuscript. 

line 73: A separate methodology section is required before results.

“The structure should include an Abstract, Keywords, Introduction, Results, Discussion, Materials and Methods, and Conclusions (optional) sections” (please see the Microsoft Word Template https://www.mdpi.com/journal/molecules/instructions)

line 78: reisolated….re-isolated

Modified

line 88: All the cultures were performed in triplicate. Sentence needs revision

According to the referee comment, this sentence was modified.

line 108, 241, 245, 258: H2O, Na2SO4, MeOH-H2O,  H2SO4…. Please correct as all numeric values within the formulae as subscripts

As suggested, we carefully checked all numeric values in the manuscript and eventually made corrections.

line 219: Mendeleum Institute of Genetics…Is it Mendeleum-Institute of Genetics and Plant Breeding, Brno, Czech Republic?

Modified

line 232: an equal number of shoots inoculated with sterile PDA plugs…..state the number of shoots please

According to the referee comment, we added the number of shoots in the manuscript (see line 241)

table 2 : numerical numbers in the table should be subscripts

As suggested, we carefully checked all numeric values in the table 2 and eventually made corrections.

conclusion: elaborate and include some future prospects please

As suggested, we elaborate and modified the conclusion section including future prospects.

We wish to thank the referee for his/her useful suggestions which improved our manuscript.

Reviewer 3 Report

Diplodia corticola is one of the most aggressive fungal pathogens for Quercus species in Mediterranean. To determine the vital factor for the virulence of D. corticola, the composition of secondary metabolites of three strains of D. corticola was comprehensively detected by metabolomic analyses. It is a very interesting research topic. The authors reported that sphaeropsidins are the major phytotoxi compounds of this fungal pathogen only via comparatively analyses on the concentration of this metabolite across different strains. However, the authors did not present any experimental evidence to support this conclusion. if sphaeropsidins was introduced into oak trees alone and led to the similar pathogenic symptoms with D. corticola, the major results could be safely supported. Furthermore, there was not any statistical values that could be found in this manuscript. Consequently, this manuscript is not recommended for being considered for publication until more solid evidence was presented. 

Author Response

Diplodia corticola is one of the most aggressive fungal pathogens for Quercus species in Mediterranean. To determine the vital factor for the virulence of D. corticola, the composition of secondary metabolites of three strains of D. corticola was comprehensively detected by metabolomic analyses. It is a very interesting research topic. The authors reported that sphaeropsidins are the major phytotoxi compounds of this fungal pathogen only via comparatively analyses on the concentration of this metabolite across different strains. However, the authors did not present any experimental evidence to support this conclusion. if sphaeropsidins was introduced into oak trees alone and led to the similar pathogenic symptoms with D. corticola, the major results could be safely supported. Furthermore, there was not any statistical values that could be found in this manuscript. Consequently, this manuscript is not recommended for being considered for publication until more solid evidence was presented.

We wish to thank the referee for the comments.

There is an extensive literature on the phytotoxicity of sphaeropsidins which was tested on host and non-host plants (see references in the manuscript). In particular, since 1996 sphaeropsidin A is recognized as a non-selective phytotoxin (see Evidente, A., Sparapano, L., Motta, A., Giordano, F., Fierro, O. and Frisullo, S., 1996. A phytotoxic pimarane diterpene of Sphaeropsis sapinea f. sp. cupressi, the pathogen of a canker disease of cypress. Phytochemistry42(6), pp.1541-1546.). However, we cannot reproduce the disease symptoms introducing sphearopsidins into oak cuttings or leaves because these systems are not assimilable to the normal vegetation conditions.

Concerning the statistical issue, Two‐way analysis of variance (ANOVA), followed by a Tukey multiple comparison test, was used to determine the statistical significance of the data reported in figure 2 and 4.